# Toxic Effects of Bisphenol A and Bisphenol S on *Chlorella Pyrenoidosa* under Single and Combined Action

**DOI:** 10.3390/ijerph19074245

**Published:** 2022-04-02

**Authors:** Junrong Li, Yingjun Wang, Na Li, Yan He, Hong Xiao, Dexin Fang, Chao Chen

**Affiliations:** Department of Environmental Engineering, College of Environment, Sichuan Agricultural University, Chengdu 611100, China; leelijunrong@163.com (J.L.); leonaqiqi1011@163.com (N.L.); heyan@sicau.edu.cn (Y.H.); xiaohong@sicau.edu.cn (H.X.); fdx@sicau.edu.cn (D.F.); chaochenhc@163.com (C.C.)

**Keywords:** bisphenol A, bisphenol S, *Chlorella pyrenoidosa*, single toxic effect, combined toxic effect

## Abstract

Bisphenol A (BPA) is an important industrial chemical; bisphenol S (BPS) is a substitute for BPA. Both are frequently detected in rivers, sewage, and surface water, and have a great impact on the water environment. The effects of BPA and BPS on cell growth, chlorophyll a content, and oxidative stress of *Chlorella pyrenoidosa* (*C. pyrenoidosa*) were studied. When BPA and BPS acted alone or in combination, compared with the blank control group, the growth of *C. pyrenoidosa* in the experimental group showed a pattern of “low promotion and high inhibition”, and the inhibition rate reached the maximum on the 6th day. Under the combined action, the reactive oxygen species (ROS) level of *C. pyrenoidosa* first increased, and then decreased. In addition, the activity of superoxide dismutase (SOD) and peroxidase (POD) increased with the increase in combined concentration. In the 0.5 P treatment group, SOD and POD activity reached peak values of 29.59 U/mg∙prot and 1.35 U/mg∙prot, respectively. The combined toxicity of BPA and BPS to *C. pyrenoidosa* was evaluated as a synergistic effect by using toxicity unit and additive index methods. This study evaluated the effects of BPA and BPS on algae in the aquatic environment, providing some data support for their potential ecological risks.

## 1. Introduction

Studies have shown that certain pollutants can accumulate in aquatic ecosystems and have serious impacts on their stability [1]. Bisphenol compounds (such as bisphenol A (BPA) and bisphenol S (BPS)) are widely used in the production of products such as electronic equipment, food packaging [2], dental sealants, baby bottles [3,4], polycarbonate plastics, and thermal plastics [5]. Due to their moderate water solubility and low vapor pressure, bisphenol compounds can be released into the environment through leachate from sewage treatment plants, sewage sludge, and landfills [5], resulting in their widespread detection in air, water, soil, sediment, waste sludge, indoor dust, human tissues, etc. Bisphenol compounds are endocrine disruptors with acute toxicity, genotoxicity, estrogenic activity, and refractory degradation properties. BPA has been reported to be detected in surface water and river sediments around the world; BPA was also found in Taihu Lake, Liaohe River, and Hunhe River in Jiaxing, China, with concentrations of 4.2–14 ng/L, 5.9–141 ng/L, and 4.4–107 ng/L, respectively [6]. The concentration of BPA in sludge dry weight is 600 ng/g dry weight [7,8]. Due to restrictions on the production and use of BPA, BPS has become one of the main substitutes for BPA [5,8]. BPS is ubiquitous in the environment. The concentration of BPS detected in thermal receipt paper samples from the United States, Japan, South Korea, and Vietnam was up to 22.0 mg/g [9]. BPS was also found in different categories of food collected from nine cities in China, with an average concentration of 0.287 ng/g fresh weight [5,8]. In South Korea and the United States, the average concentration of BPS in sludge samples from sewage treatment plants is 44.9 ng/g dry weight and 34.5 ng/g dry weight, respectively [5,8]. In Taihu Lake of China and the Attiya River of India [10], the maximum concentrations of BPS were 0.016 μg/L and 7.2 μg/L, respectively. Therefore, it is necessary to evaluate the effects of BPA and BPS on humans and the environment.

The physiological and ecological effects of BPA and BPS have been well documented. BPA may promote obesity by inhibiting the neural circuitry that regulates feeding behavior or changes adipocyte differentiation [11]. Aker et al. [12] evaluated the correlation between BPS concentrations in the urine of pregnant women and thyroid hormone levels throughout the whole pregnancy, and their findings indicated that BPS was related to thyroid-stimulating hormone and thyroid hormone. Studies have shown that low doses of BPA can promote the growth of aboveground plant organs, while low concentrations of BPA (1.5 mg/L) can increase plant height, stem dry weight, fresh leaf dry weight, and soybean leaf area [13]. Studies have shown that BPA has a great impact on the growth of *Platymonas helgolandica*, and the 96 h EC_50_ is 9.32 mg/L, making it a highly toxic substance [14]. BPA also affects the photosynthetic system of algae to varying degrees [15,16]. Gattullo et al. [17] found that BPA exposure disrupts photosynthesis in photosynthetic system II (PS II) in *Monoraphidium braunii*, thereby reducing the net efficiency of Chl-a and PS II. Under normal circumstances, algae maintain a dynamic balance of reactive oxygen species (ROS) levels through antioxidant systems. When microalgae are exposed to a polluted environment, a large number of ROS will be produced and accumulated on the chlorophyll of algae cells, leading to the destruction of internal balance, abnormal function, and even the death of cells [18,19]. Studies have shown that exposure of *C. caspia* to BPA (4–6 mg/L) can enhance the activity of SOD, thus effectively eliminating ROS [20]. In addition, many scholars have also carried out relevant studies on the combined toxicity of mixed pollutants. For example, Nong Qiongyuan et al. [21] studied the combined toxicity of the mixture of antibiotics and triazole fungicides on *Selenastrum capricornutum*, and found that the combined toxicity of the mixture was different at different concentrations. Due to their widespread distribution, BPA and BPS may coexist in the marine environment. So far, most studies have focused on the effects of BPA on algae, but the effects of BPA and BPS on algae are still scarce. However, we do not know whether the combination of BPA and BPS affects algae.

The current high detection rates and concentrations of BPA and BPS in rivers raise legitimate concerns about the impact on aquatic biota. Algae are the most important primary producers in aquatic ecosystems [22], and are also an important indicator for monitoring and evaluating water’s environmental quality [23]. Numerous studies have found that with increasing amounts of BPA and BPS released into the water, they can accumulate in algal cells and throughout the aquatic food chain, which can have negative effects on aquatic ecosystems. Therefore, in this study, *C. pyrenoidosa* was selected as the research object to evaluate the single and combined toxic effects of BPA and BPS. The cell numbers, chlorophyll a contents, and ROS levels of algae were determined, and the toxicity mechanism was discussed. This study provides ideas for the combined risk assessment of BPA and BPS in aquatic ecosystems.

## 2. Materials and Methods

### 2.1. Drug and Microalgal Culture

The purity of BPA and BPS was greater than 99.0%, purchased from Shanghai Yien Chemical Reagent Co., Ltd. We adjusted the pH value of the prepared PBS solution to 7.4 [24], and cooled it for later use after sterilization. *Chlorella pyrenoidosa* (FACHB-27) was purchased from Institute of Hydrobiology, Chinese Academy of Sciences. The *C. pyrenoidosa* was transferred with BG11 and placed in a biochemical incubator with a 12 h:12 h light:dark cycle, temperature of 25 °C, and light intensity of 10,000~14,000 lx.

### 2.2. Bisphenol A and Bisphenol S Mother Fluids

First, 0.1 g of BPA (BPS) was dissolved with 0.5 mL of dimethyl sulfoxide solution, and then we added BG11 medium to reach a volume of 500 mL, prepared into 200 mg/L mother liquor. The pH was adjusted to 7.1 with 1 mol/L NaOH, and then autoclave sterilization was carried out after the temperature was reduced.

### 2.3. Median Effective Concentrations of BPA and BPS (EC_50_)

In the EC_50_ experiment for BPA, there were 9 concentration gradients of 0, 2, 5, 8, 11, 15, 20, 25, and 40 mg/L. In the EC_50_ experiment for BPS, 9 concentration gradients of 0, 5, 10, 15, 20, 40, 50, 60, and 70 mg/L were set. Samples were taken on the 6th day, the density of algal cells was measured, and the growth inhibition rate was calculated compared with the blank control. The logarithmic value of the tested substances was the abscissa, and the inhibition rate of growth was the ordinate. The 144 h EC_50_ values of BPA and BPS were obtained via Origin software fitting.

### 2.4. Single Toxicity Exposure Experiments for BPA and BPS

The experiment period was 6 days, with 1 blank group and 5 experimental groups. In the experiment, *C. pyrenoidosa* liquid was used in the logarithmic growth phase, and the initial density of *C. pyrenoidosa* was 2.0 × 10^5^ cells/mL; then, the mother solution of BPA was added and placed in 250 mL triangular vials. The culture volume was 170 mL, and the concentrations of BPA were 2.0, 5.0, 8.0, 11.0, and 15.0 mg/L. It was verified that BPA had a toxic effect on algal species under the concentration system. Then, they were exposed for 6 days in a biochemical incubator with illumination from 14,000 to 17,000 lx and a light:dark cycle of 12 h:12 h. These steps were repeated 3 times for each test concentration. The position of the triangular bottles in the illumination incubator was changed regularly every day to ensure that all of the bottles received uniform light. After the exposure experiment, the cell density and chlorophyll a content of the algae liquid were measured regularly every day. The contents of MDA, POD, SOD, and BPA in the culture medium were determined on the 6th day.

The concentration of BPS in *C. pyrenoidosa* was set as 5.0, 10.0, 15.0, 20.0, and 40.0 mg/L. The other specific operation and determination indices were the same as in the BPA single toxicity exposure experiment.

### 2.5. Combined Toxicity Exposure Experiment for BPA and BPS

Based on the 144 h EC_50_ of BPA and BPS on *C. pyrenoidosa*, a single-compound 144 h EC_50_ was defined as a toxicity unit (1 TU), and the mixed system was defined as P according to the toxicity unit ratio of 1:1. The mixing system was set with 0.05 P, 0.1 P, 0.2 P, 0.3 P, 0.4 P, and 0.5 P concentrations. The concentrations of BPA and BPS in the mixing system are shown in Table 1.

In the experiment, the final culture volume was 300 mL, so that the concentration of the mixed system reached 0 P, 0.05 P, 0.1 P, 0.2 P, 0.3 P, 0.4 P, and 0.5 P. The other specific operation and determination indices were the same as in the BPA single toxicity exposure experiment.

### 2.6. Determination of Chlorophyll a Content

The extraction and determination of chlorophyll followed the method of Feng Qingying et al. [25], with some improvements. The stressed *C. pyrenoidosa* sample was centrifuged at 5000 r/min for 10 min, and then the supernatant was abandoned and 5 mL of 90% acetone was added. The samples were stored in the refrigerator overnight in the dark to complete the chlorophyll extraction, and then centrifuged again for 5 min to collect the supernatant. Finally, the optical density was measured at 630 nm, 645 nm, 663 nm, and 750 nm, using 90% acetone as a blank. No turbidity was observed during the experiment. Chlorophyll a content was calculated by the following method:$$C(\mathrm{mg}/m^{3})=[11.64\;\times \;(A_{663}\;-\;A_{750})\;-\;2.16\;\times \;(A_{645}\;-\;A_{750})+0.10\;\times \;(A_{630}\;-\;A_{750})]\cdot v_{1}/v$$
where v_1_ and v are the volume of the extract after constant volume and sample volume, respectively.

### 2.7. Antioxidant System Analysis

#### 2.7.1. Extraction of Crude Enzyme from Alga Cells

A 20 mL sample of algal liquid was taken and centrifuged at 4500 r/min for 10 min in a low-speed centrifuge. The supernatant was poured out and washed twice with 0.1 mol/L PBS suspension at pH 7.4. The culture medium attached to *C. pyrenoidosa* was removed and then centrifuged to collect algal cells, and liquid nitrogen was poured into the centrifuge tube. After the liquid nitrogen was evaporated, the frozen algal cells were transferred to the homogenizer for ice bath grinding. After 3–4 min, 600 μL of PBS buffer solution was added. The broken algal cells were transferred to a 3 mL Eppendorf tube and centrifuged at 10,000 r/min in a high-speed centrifuge for 5 min. The supernatant was used for the determination of enzyme activity.

#### 2.7.2. Assays of Oxidative Stress and Antioxidant Enzyme Activities

We used 2′,7′-dichlorodihydrofluorescein diacetate (DCFH-DA) to detect the ROS levels of algal cells [26]. The basic principle is that DCFH-DA itself does not fluoresce, so it can freely cross the cell membrane. After entering the cell, DCFH can be hydrolyzed by intracellular esterase to form DCFH, and DCFH does not penetrate the cell membrane, so probes can be easily accumulated in the cell. Intracellular reactive oxygen species can oxidize non-fluorescent DCFH to produce fluorescent DCF. The intensity of green fluorescence is proportional to the level of reactive oxygen species.

Algal cells were collected by centrifugation and washed twice with PBS [27]. Then, the diluted DCFH-DA solution was added to the test tube and incubated in a dark box at 25 °C for 30 min. The contents of MDA, SOD, and POD were determined with detection kits (Nanjing Jiancheng Institute of Biological Engineering, Nanjing, China). MDA, SOD, and POD were determined by the thiobarbituric acid (TBA) method, hydroxylamine method, and guaiacol method respectively, and the specific determination steps were carried out according to the requirements of the kit.

### 2.8. Combined Toxic Effect Evaluation

The toxicity unit method and additive index method were used to evaluate the combined toxicity effect of BPA and BPS. In this experiment, the 144 h EC_50_ values of BPA and BPS to *C. pyrenoidosa* under the single and combined toxic effects were obtained, and the value of S was calculated by substituting the following formula:$$S=\frac{A_{m}}{A_{1}}+\frac{B_{m}}{B_{1}}+\frac{C_{m}}{C_{1}}+\cdots \cdots$$
where S is the sum of the additive effects of biological toxicity; A_1_, B_1_, and C_1_ are the EC_50_ values of A, B, and C, respectively; and A_m_, B_m_, and C_m_ are the EC_50_ values of each poison in the toxicity of the mixture.
(1)$$S\;\leq \;1\;:\;\mathrm{AI}=\left(\frac{1}{S}\right)-\;1.0\;S>1\;:\;\mathrm{AI}=1.0-\;S$$

Evaluation criteria of the combined effects of the additive index (AI) on poisons:

When AI = 0, AI > 0, and AI < 0, the combined toxic effects are simple summation, synergism, and antagonism, respectively.

### 2.9. The Data Processing

All data in this study were calculated using Excel 2010(Microsoft Corporation, Beijing, China) for results and Origin 2018 for plots. Statistical analysis was applied and one-way ANOVA was conducted using SPSS 20.0 statistical software (International Business Machines Corporation, Armonk, New York, NY, USA). The results showed that *p* < 0.05, indicating significant difference between the two groups.

## 3. Results and Discussion

### 3.1. Effects on Cell Density

As can be seen from Figure 1A, on the 1st day, the density of algal cells in the 2 mg/L treatment group was 19% higher than that in the control group, showing a promoting effect, while the other treatment groups showed an inhibitory effect. Low concentrations of BPA could stimulate the division of algal cells, and finally showed a promoting effect. On the 4th day, the 2 mg/L treatment group changed from stimulation to inhibition, and the inhibition rates of the 5, 8, and 11 mg/L treatment groups were 7%, 14%, and 21%, showing an increasing trend. It may be that with the increase in BPA stress time, the damage to algal cells is further intensified, or it may be that high concentrations of BPA inhibit the synthesis of chlorophyll a and cause varying degrees of damage to the photosynthetic systems of algal cells, leading to a decrease in algal cell density. This is similar to the findings of Xiang et al. [28]. On the 6th day, the inhibitory effect of BPA on *C. pyrenoidosa* decreased in the 11 mg/L and 15 mg/L treatment groups. Studies have shown that algae have the ability to degrade phenols given the presence of a sufficient carbon source [29]. BPA is a derivative of phenolic substances; this may be why in this study, on the sixth day, the inhibitory effect of BPA on algal cells decreased significantly. The effects of BPS on the growth of *C. pyrenoidosa* conform to the rule that low concentration promotes high concentration inhibition (Figure 1B). On the 2nd day, the difference between the 5, 20, and 40 mg/L treatment groups and the control group was significant (*p* < 0.05). The density of algal cells in the 5 mg/L treatment group was 0.5% higher than that in the blank group, showing that BPS promoted growth. On the 4th day, the inhibition rate of 15 mg/L to algal cells increased to 10%; in the 20 mg/L and 40 mg/L groups the rate was 20% and 28%, respectively. It can be seen that with the extension of culture time, the adaptability of algae cells to the environment is enhanced, and *C. pyrenoidosa* can resist the damage caused by a certain concentration of BPS through self-regulation. On the 6th day, the inhibition rates of 15, 20, and 40 mg/L reached their peaks, which were 13%, 22%, and 32%, respectively.

The combined effect of BPA and BPS on algal cell density was similar to that of the single effects (Figure 1C). On the 1st day, the density of algal cells in the 0.05 P treatment group was 4% higher than that in the control group, showing a promoting effect, while the residual treatment group showed different degrees of inhibition effect. On the 5th day, the inhibitory effect of different treatment groups on *C. pyrenoidosa* was intensified, and the inhibition rates of the 0.1 P, 0.2 P, 0.3 P, 0.4 P, and 0.5 P treatment groups were 5%, 18%, 32%, 48%, and 61%, respectively. Under the single action of BPA and BPS, the inhibition rate of algal cells decreased on the 5th day, but the effect was opposite in the combined effect. This indicates that the combined concentration caused serious damage to the internal structure of *Chlorella* cells, which could not be repaired by self-regulation. It can also be seen that the inhibition of algal cell density is more serious under the combined effect.

### 3.2. Evaluation of the Toxicity Effects of BPA and BPS on C. pyrenoidosa

On the 6th day, BPA was 3.7 times more toxic than BPS (Figure 2). Table 2 shows the toxic effects of BPA and BPS at different concentrations on *C. pyrenoidosa* at a 1:1 toxicity ratio. The results showed that the order of toxicity was combined toxicity > single toxicity, BPA > BPS. According to the algal growth toxicity standard, when it is higher than 10 mg/L and lower than 100 mg/L, the toxicity is moderate. Therefore, the single toxicity of BPA and BPS to *C. pyrenoidosa* was toxic, while the combined toxicity was highly toxic. Meanwhile, it can be seen from the table that the combined toxicity of BPA and BPS to *C. pyrenoidosa* was synergistic.

### 3.3. Effects on Chlorophyll A

As can be seen from Figure 3A, there were no significant differences between any of the groups and the blank group on the 1st day (*p* > 0.05). On the 2nd day, the content of chlorophyll a in the 2 mg/L treatment group was significantly higher than that in the blank group, by 13.7%, indicating that the 2 mg/L treatment group stimulated the synthesis of chlorophyll a in *C. pyrenoidosa*. On the 6th day, compared with the control group, the inhibition rates of chlorophyll a concentration were 11%, 20%, 42%, and 46% at 2, 5, 8 and 11 mg/L, respectively, and the inhibition rate of chlorophyll a content reached the peak (61%) at 15 mg/L. Studies have shown that the decrease in chlorophyll a may be because BPA can react with bioactive groups in algae, thus inhibiting many physiological processes, including cell division and photosynthesis [30,31]. As we can see from Figure 3B, on the 3rd day, there were significant differences between all treatment groups and the blank group (*p* < 0.05). The inhibition of chlorophyll a in the 10, 15, and 40 mg/L groups reached the maximum value, which was 12%, 24%, and 59%, respectively. On the 4th day, the inhibition rate of the *C. pyrenoidosa* by BPS decreased to 4% in the 10 mg/L treatment group, because with the increase in culture time, the adaptability of *C. pyrenoidosa* to the environment increased, thus reducing the stress caused to it by BPS. On the 6th day, the chlorophyll a content of the 5 mg/L treatment group showed a stimulant effect, but the inhibition effect on chlorophyll a content was not further aggravated. The inhibition of photosynthesis may have been due to the increase in the concentration of BPS and the attack of the cells—that is, the production of excessive ROS, which attack the chlorophyll and chloroplast membranes [32,33], affecting the photosynthetic efficiency of *C. pyrenoidosa*.

On the 1st day, chlorophyll a content in the 0.05 P, 0.1 P, 0.2 P, and 0.3 P treatment groups was higher than that in the control group, showing a slight stimulant effect. On the 2nd day, the 0.05 P group still showed a stimulant effect, while the 0.1 P, 0.2 P, and 0.3 P groups changed from a mild stimulant effect to an inhibitory effect (Figure 3C). On the 6th day, the inhibition rate of the 0.2 P, 0.3 P, 0.4 P, and 0.5 P groups reached the maximum of 39%, 44%, 61%, and 85%, respectively. It can be inferred that the combined effect of BPA and BPS on the photosynthetic system of algal cells is greater than the single effects of BPA and BPS. It may be that the excessive accumulation of reactive oxygen species destroys the thylakoid membrane and chloroplast structure and pigment under the combined action [34]. This conclusion is consistent with the results of the combined toxicity assessment, which was described as synergistic.

### 3.4. Oxidative Stress

#### 3.4.1. Analysis of ROS Levels

Excessive ROS can cause lipid peroxidation in algal cells. As the main product of lipid peroxidation, ROS levels increased compared with the control group (Figure 4). Among all of the BPA test groups, ROS levels in the 11 mg/L treatment group were the highest, and were 1.5 times higher than those in the control group. Therefore, it can be said that high concentrations of BPA induced excessive ROS in the tested algal cells. For the BPS experimental groups, ROS levels in the two low-concentration treatment groups showed no significant differences compared with the control group, but ROS levels gradually increased with the increase in concentration. When BPA and BPS were mixed at a toxicity ratio of 1:1, ROS levels changed, showing a trend of first increasing and then decreasing; this may be because the algal cells were severely damaged by peroxide at 0.05 P and 0.1 P, so ROS levels increased significantly compared with the control group.

#### 3.4.2. Analysis of MDA Content, and POD and SOD Activity

MDA content can be used as an indicator of the degree of lipid peroxidation and severity of cell damage [35]. In addition, MDA can also lead to degradation of chlorophyll, thus affecting photosynthesis [36]. Simultaneously, SOD can scavenge ROS to maintain the balance of oxidation and anti-oxidation in algae. SOD and POD are important antioxidant enzymes in organisms. Once the ROS precursor produces O^2−^, SOD plays an important role in eliminating O^2−^ to reduce cell damage. It catalyzes the deformation of O^2−^ to reduce and prevent lipid peroxidation. POD is a kind of oxidoreductase, which can decompose H_2_O_2_ in algal cells, so as to reduce and prevent the toxic effect of H_2_O_2_ on algal cells. In the present study, the intracellular ROS level, MDA content, and POD and SOD activity were measured to assess the extent of oxidative stress.

MDA content first gradually increased and then decreased with the increase in BPA treatment concentration (Figure 5A). In the 2 mg/L treatment group, there was no significant difference compared with the blank group (*p* > 0.05), indicating that BPA at this concentration did not cause serious oxidative damage to *C. pyrenoidosa*, the function of the antioxidant enzyme system was strengthened, and various active oxygen substances in various cells were cleared by the algal cells in time, so that the physiological metabolic level of the cells returned to normal. The cells did not suffer serious oxidative damage. SOD activity first increased and then decreased with the increase in BPA concentration. The maximum SOD activity in the 11 mg/L treatment group was 31.46 U/mg∙prot—2.4 times of that in the control group. In the 15 mg/L treatment group, SOD activity decreased. The increase in SOD activity may have been due to the protection strategy for the increase in ROS production and the adjustment in response to the oxidation conditions [37]. When SOD activity is inhibited, it may be because high concentrations of BPA seriously interfere with the normal function of the antioxidant enzyme system, or it may be because ROS accumulation in *C. pyrenoidosa* exceeds SOD clearance capacity, resulting in inhibition of SOD activity. POD activity showed an upward trend. There was no significant difference in POD activity between the 2 mg/L treatment group and the blank group (*p* > 0.05). With the increase in BPA concentration, POD activity began to increase significantly (*p* < 0.05), and the POD content increased to the maximum at 15 mg/L, which was 1.13 U/mg∙prot. On the 6th day, the MDA value showed an upward trend as the concentration of the treatment group increased (Figure 5B). The 5 mg/L and 10 mg/L treatment groups had significant differences compared with the blank group (*p* < 0.05), but no significant difference between the groups (*p* > 0.05). This indicates that the oxidative damage caused by BPS to *C. pyrenoidosa* was relatively light at the 0–10 mg/L concentrations of BPS, and that the algal cells had not suffered serious oxidative damage thanks to timely removal of various intracellular reactive oxygen substances. SOD and POD activities increased with the increase in BPS concentration, and there were no significant differences between the 10, 15, and 20 mg/L treatment groups (*p* > 0.05). In the 40 mg/L treatment group, SOD activity reached its peak, which was 44.08 U/mg∙prot—3.5 times that of the control group. The POD activity of the experimental groups showed an upward trend. POD content in the 5 mg/L and 10 mg/L treatment groups was not significantly different from that in the blank group (*p* > 0.05). This indicates that POD activity was not affected at these concentrations. In the 15, 20, and 40 mg/L treatment groups, POD content began to increase significantly (*p* < 0.05), and reached the maximum value of 1.54 U/mg∙prot at 40 mg/L, indicating that the cells were suffering serious oxidative damage at this time. The normal physiology of the algal cells was destroyed.

There was a difference between joint action and single action; the MDA value showed a trend of first increasing and then decreasing (Figure 5C). In the 0.05 P treatment group, the MDA value reached its highest value of 1.52 nmol/mg∙prot, which was 1.6 times that of the blank group. With the increase in BPA and BPS concentrations, MDA content in the 0.3 P, 0.4 P, and 0.5 P treatment groups had no significant change (*p* > 0.05), with values 15%, 15%, and 2% lower than that in blank group, respectively. In this experiment, there may be two reasons for the increase and then decrease in MDA content: One is that the algal cells were subjected to severe oxidative stress at 0.05 P and 0.1 P, which led to the increase in MDA content. With the further increase in BPA and BPS concentrations, the cell membrane structure was seriously damaged, causing serious damage to the algal cells. Therefore, MDA content showed a downward trend. The other reason is that at 0.05 P and 0.1 P, algal cells were slightly damaged by peroxide, and MDA content increased. The antioxidant system in the algal cells cleared free radicals in time, so the MDA content decreased. The combined toxicity of BPA and BPS is due to the excessive accumulation of ROS through oxidative stress, which affects cell membrane structure and inhibits cell growth. The activities of SOD and POD increased gradually with the increase in the combined concentration of BPA and BPS. In the 0.5 P treatment group, SOD and POD activities reached peak values of 29.59 U/mg∙prot and 1.35 U/mg∙prot, respectively. In the combined toxicity test, all experimental groups followed the results of the combined toxicity evaluation; that is, the combined effect of BPA and BPS on the oxidative stress of *C. pyrenoidosa* was greater than the single effects.

## 4. Conclusions

In this study, we investigated the toxic effects of BPA and BPS on *C. pyrenoidosa*, including inhibition of photosynthesis and oxidative damage. The results showed that BPS induced oxidative stress and significantly inhibited photosynthesis. Low concentrations of BPA promoted the synthesis of chlorophyll a, suggesting that oxidative damage might be responsible for inhibiting cell growth. For BPS, high concentrations of BPS not only inhibited growth and photosynthesis, but also caused serious oxidative damage, indicating that the growth inhibition is mainly caused by BPS-induced oxidative damage and inhibition of photosynthesis. The combined toxicity of BPA and BPS is synergistic. These findings are valuable for improving our understanding of the toxic effects of BPA and BPS on marine microalgae, as well as for assessing the actual environmental pollution of other pollutants and their substitutes.

## Figures and Tables

**Figure 1 ijerph-19-04245-f001:**
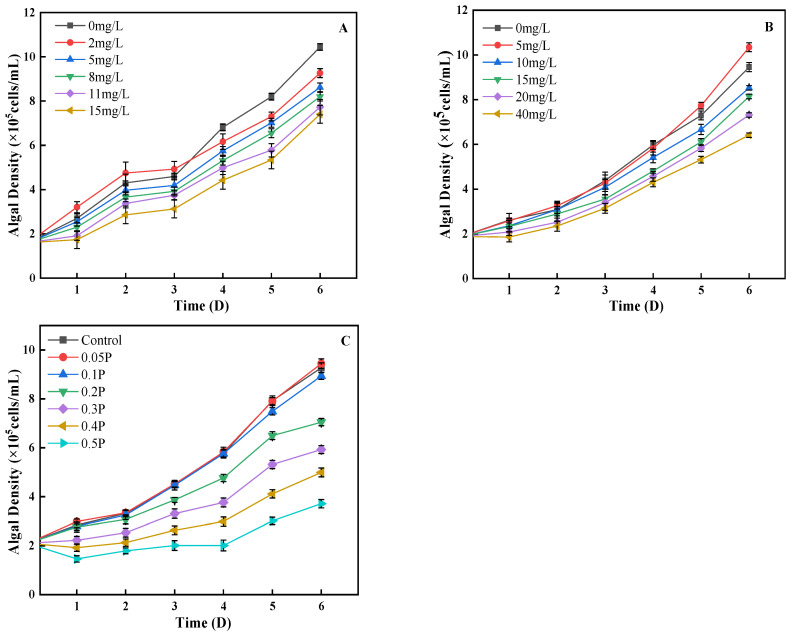
Effects of BPA, BPS, and BPA + BPS on the algal density of *C. pyrenoidosa*: (**A**,**B**) show the single effects of BPA and BPS on the algal density; (**C**) shows the effects of joint BPA and BPS toxicity on the algal density.

**Figure 2 ijerph-19-04245-f002:**
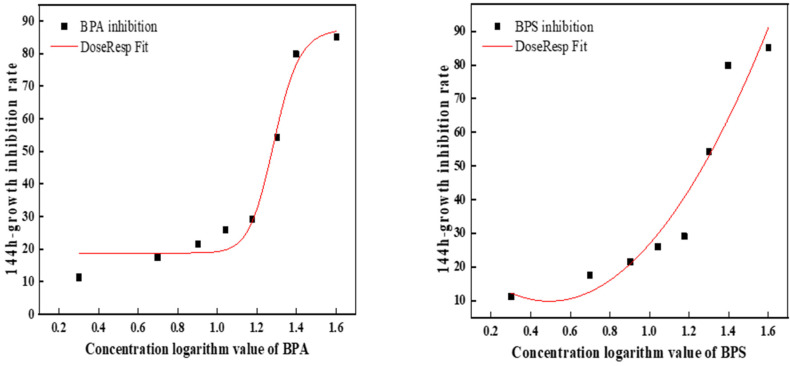
The fitted curve of the 144 h EC_50_ of BPA and BPS.

**Figure 3 ijerph-19-04245-f003:**
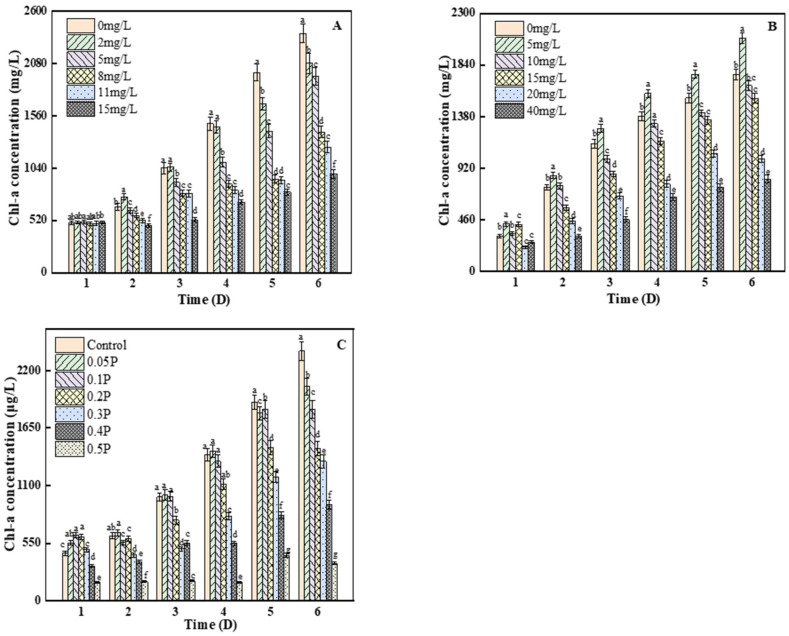
Effects of BPA, BPS, and BPA + BPS on Chl-a of *C. pyrenoidosa*: (**A**,**B**) show the effects of single BPA and BPS on Chl-a content; (**C**) shows the effects of joint BPA and BPS toxicity on Chl-a content. Note: Data are means ± SD of the three replicates. Different lowercase letters indicate significant differences between the treatments (*p* < 0.05).

**Figure 4 ijerph-19-04245-f004:**
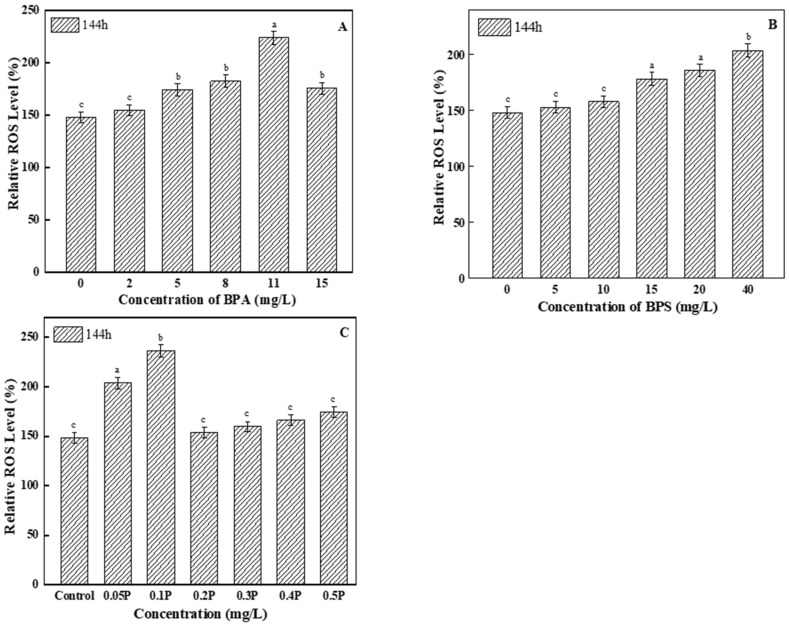
Effects of BPA, BPS, and BPA + BPS on the ROS levels of *C. pyrenoidosa*: (**A**,**B**) show the effects of single BPA and BPS on ROS levels; (**C**) shows the effects of joint BPA and BPS toxicity on ROS levels. Note: Data are means ± SD of the three replicates. Different lowercase letters indicate significant differences between the treatments (*p* < 0.05).

**Figure 5 ijerph-19-04245-f005:**
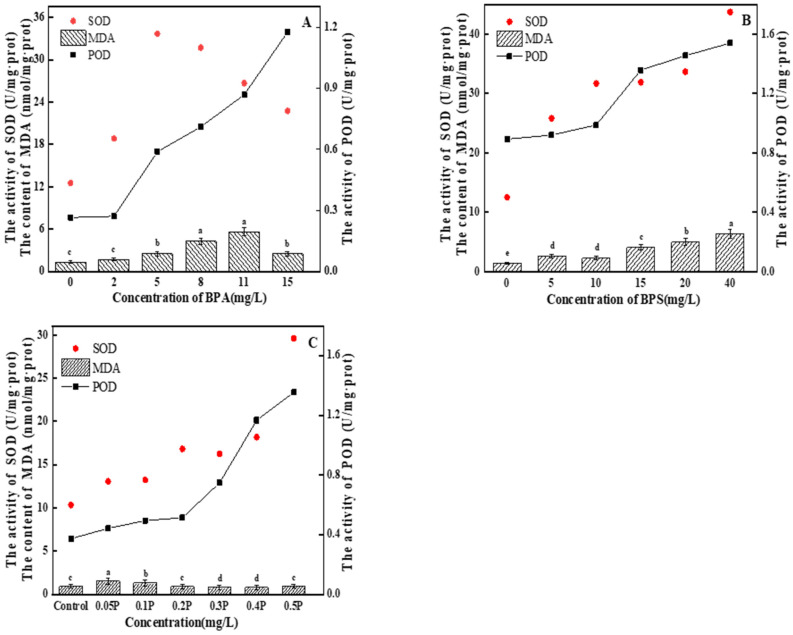
(**A**–**C**) represent the effects of BPA, BPS, and BPA + BPS, respectively, on the intracellular MDA content, and the POD and SOD activity, in *C. pyrenoidosa*. Note: Data are means ± SD of the three replicates. Different lowercase letters indicate significant differences between the treatments (*p* < 0.05).

**Table 1 ijerph-19-04245-t001:** Concentration combinations of BPA and BPS under 1:1 toxicity.

Toxicity Ratio (BPA:BPS)		Concentration Combination (mg/L)						
		0 P	0.05 P	0.1 P	0.2 P	0.3 P	0.4 P	0.5 P
1:1	BPA	0	0.27	0.53	1.06	1.59	2.12	2.66
	BPS	0	0.99	1.98	3.96	5.94	7.92	9.90

**Table 2 ijerph-19-04245-t002:** Adjustment results of the 144 h joint toxicity test.

	The Single Toxicity of 144h-EC_50_ (mg/L)	The Joint Toxicity of 144h-EC_50_ (mg/L)	S	AI	Conclusion
BPA	17.7	4.8	0.543	0.842	Synergy
BPS	66.07	18			

## Data Availability

Not applicable.

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
