# Peer review of "Toxic Effects of Bisphenol A and Bisphenol S on Chlorella Pyrenoidosa under Single and Combined Action"

_ijerph, 2022, doi:10.3390/ijerph19074245_

Round 1
Reviewer 1 Report
As mentioned in the Introduction, the toxicity of BPA and BPS has been revealed in previous studies and in the present study the authors aim to evaluate their combined effects. However, the credibility of the mixture assessment is of question as explained in the comments on the Materials and Method section. It is possible to evaluate the type of combined effects by comparing the toxicity of mixture with the additive effect (for example, based on the concept of concentration addition). Yet, with the description in the manuscript, the analysis is doubtful. That analysis was implemented based on the growth of algae. Comparison in the responses related to oxidative stress between single and mixture exposure should also be conducted.
Abstract:
Line 13: The species name should be given in full at the first time.
Lines 13, 14, 15, 17: The species name should be written in italic.
Lines 13 – 14: “When BPA and BPS were single and combined exposure” is a poor expression.
Lines 14 – 16: What are ROS, SOD, and POD? Their full name should be introduced first. Moreover, the biomarker responses should be compared with those of unexposed algae.
Lines 18 – 20: It is not suitable to mention the objectives at the end of the Abstract. Here the authors should mention the potential implication of their findings in a broader term.
Introduction:
Line 31: What is “acute toxicity”?
Lines 32 – 33: “BPA has been reported… around the world”: this sentence is quite redundant and not informative.
Lines 33 – 44: These levels are much much higher than the concentrations studied by the authors. So a reasonable argument for the investigated concentration is required in the Materials and Method section.
Materials and Methods: The most important issue of the work is to evaluate the combined toxicity of BPA and BPS. However, the description of the analysis (lines 157 – 168) is confusing. With the description here, the method is not accurate. The type of combined effects could be evaluated based on the toxicity of the mixture compared with the additive effect, in the present study based on the concept of concentration addition. In the method, the authors just simply calculated the sum of toxic units based on toxicity of single substances. Where is this compared with combined toxicity? Another concern is related to the concentrations investigated. They are remarkably higher than the levels that are usually reported. Then why were these concentrations used?
Line 94: EC50 is usually defined as the median effect concentration, not semi-maximum effect concentration.
Lines 94 – 117: The separation of these two sections causes confusion.
Lines 129 – 135: The description of the analysis of chlorophyll a content is too superficial. The equation was not explained well enough.
Results and Discussion:
Lines 176 – 182: The growth of algae was stimulated at low exposure levels. Then does it mean that at the environmentally relevant concentration, these substances are expected not to adversely affect the algae? This further adds further questions on the significance of the study.
From lines 282: The authors investigated responses in terms of oxidative stress in both single and mixture exposure. However, comparison between in these responses between the two conditions is completely lacking.
Reviewer 2 Report
Li et al. evaluated the effects of BPA and BPS on C. pyrenoidosa in terms of cell growth, chlorophyll a content and oxidative stress.
BPA and BPS are common plastic additives and are considered emergent pollutants, due to their effects as estrogenic disruptors and due to the fact that they usually end up in the environment and therefore affect human health as well as the ecosystems, papers investigating these pollutants should be treated with interest.
The title of the paper is clear, the abstract is concise but should briefly state the results.
Introduction is lacking data regarding similar studies [1-3] I suggest stating these as they have important links with the presented study.
Materials and methods
2.6. Determination of chlorophyll a content
State the method clearly, was the extraction performed under low illumination?
Was there turbidity/microbial growth observed?
2.7.2. Assays of oxidative stress and antioxidant enzyme activities
How was the Malondialdehyde (also SOD and POD) measured? Or state which kits were used.
References
- Pop, C.-E.; Draga, S.; Măciucă, R.; Niță, R.; Crăciun, N.; Wolff, R. Bisphenol A Effects in Aqueous Environment on Lemna minor. Processes 2021, 9, 1512. https://doi.org/10.3390/pr9091512
- Wang, S., Wang, L., Hua, W., Zhou, M., Wang, Q., Zhou, Q., & Huang, X. (2015). Effects of bisphenol A, an environmental endocrine disruptor, on the endogenous hormones of plants. Environmental science and pollution research international, 22(22), 17653–17662. https://doi.org/10.1007/s11356-015-4972-y
- Li, YT., Liang, Y., Li, YN. et al. Mechanisms by which Bisphenol A affect the photosynthetic apparatus in cucumber (Cucumis sativus L.) leaves. Sci Rep 8, 4253 (2018). https://doi.org/10.1038/s41598-018-22486-4
Round 2
Reviewer 2 Report
The authors have revised their work accordingly, missing data was added and I now find the manuscript suitable for publication.